# Using a Hybrid Neural Network and a Regularized Extreme Learning Machine for Human Activity Recognition with Smartphone and Smartwatch

**DOI:** 10.3390/s23063354

**Published:** 2023-03-22

**Authors:** Tan-Hsu Tan, Jyun-Yu Shih, Shing-Hong Liu, Mohammad Alkhaleefah, Yang-Lang Chang, Munkhjargal Gochoo

**Affiliations:** 1Department of Electrical Engineering, National Taipei University of Technology, Taipei 10608, Taiwan; thtan@ntut.edu.tw (T.-H.T.); jyunyu62526@gmail.com (J.-Y.S.); muhai@mail.ntut.edu.tw (M.A.); ylchang@ntut.edu.tw (Y.-L.C.); 2Department of Computer Science and Information Engineering, Chaoyang University of Technology, Taichung 413310, Taiwan; 3Department of Computer Science and Software Engineering, United Arab Emirates University, Al-Ain 15551, United Arab Emirates; mgochoo@uaeu.ac.ae

**Keywords:** mHealth, human activity recognition, bidirectional gated recurrent unit (BiGRU), regularized extreme machine learning (RELM)

## Abstract

Mobile health (mHealth) utilizes mobile devices, mobile communication techniques, and the Internet of Things (IoT) to improve not only traditional telemedicine and monitoring and alerting systems, but also fitness and medical information awareness in daily life. In the last decade, human activity recognition (HAR) has been extensively studied because of the strong correlation between people’s activities and their physical and mental health. HAR can also be used to care for elderly people in their daily lives. This study proposes an HAR system for classifying 18 types of physical activity using data from sensors embedded in smartphones and smartwatches. The recognition process consists of two parts: feature extraction and HAR. To extract features, a hybrid structure consisting of a convolutional neural network (CNN) and a bidirectional gated recurrent unit GRU (BiGRU) was used. For activity recognition, a single-hidden-layer feedforward neural network (SLFN) with a regularized extreme machine learning (RELM) algorithm was used. The experimental results show an average precision of 98.3%, recall of 98.4%, an *F*_1_*-score* of 98.4%, and accuracy of 98.3%, which results are superior to those of existing schemes.

## 1. Introduction

In 2019, the World Health Organization (WHO) proposed guidelines for digital health interventions which provide information on the potential benefits, harms, feasibility, and resources required for such interventions [1]. Digital health techniques include mobile health (mHealth) and electronic health (eHealth) and have been recognized as important tools for combating pandemic diseases [2,3]. mHealth employs mobile devices, mobile communication techniques, and the Internet of Things (IoT) to enhance healthcare in various areas, including traditional telemedicine, healthcare monitoring and alerting systems, drug-delivery programs, and medical information awareness, detection, and prevention [4,5,6].

Presently, smartphones and smartwatches are the most important mobile devices in mHealth [7,8]. They are equipped with various sensors and have many applications in the monitoring, prevention, and detection of diseases. In more advanced services, they can even provide basic diagnoses for conditions such as cardiology [9,10], diabetes [11,12], obesity [13,14], smoking cessation [15], and chronic diseases [16]. Health and fitness applications (apps), which can detect the numbers of steps walked and stairs climbed in a day using accelerometers and gyroscopes, are the most popular apps. These physical activities are used to calculate the number of calories spent. Over the past decade, recognition of physical activities has been applied to prevent falls among the elderly [17,18,19]. However, with the COVID-19 pandemic and an aging society, monitoring quarantined or elderly individuals has become a major issue in mHealth. Numerous studies have shown that people’s activities have strong correlations with their physical and mental health [20,21]. Therefore, recognizing physical activities using accelerometers and gyroscopes embedded in smartphones and smartwatches is a critical challenge in mHealth.

In recent years, deep learning (DL) and machine learning (ML) have been widely applied in mHealth [22,23,24,25]. In these studies, DL and ML models are not only used for diagnosing, estimating, mining, and delivering physiological signals, but also for preventing chronic diseases. However, in mHealth, the big data need to be delivered to servers, such as hospitals or health management centers. Therefore, telecommunications and navigation technologies are also important, in which the technologies of artificial intelligence have been applied [26,27]. Stefanova-Pavlova et al. proposed the refined generalized net (GN) to track users’ locations [28]. Silva et al. used Petri nets to process the reliability and availability of wireless sensor networks in a smart hospital [29]. Ruiz et al. proposed a tele-rehabilitation system to assist with physical rehabilitation during the COVID-19 pandemic [30].

Convolutional neural networks (CNNs) can extract features from signals, while long short-term memory (LSTM) can recognize time-sequential features. Therefore, some studies have proposed deep neural networks that combine CNNs and LSTM to recognize physical activities [31,32]. Li et al. utilized bidirectional LSTM (BiLSTM) for continuous human activity recognition (HAR) and fall detection with soft feature fusion between the signals measured by wearable sensors and radar [33]. The extreme learning machine (ELM) has shown excellent results in classification tasks with extremely fast learning speed [34]. Chen et al. proposed an ensemble ELM algorithm for HAR using smartphone sensors [35]. Their results showed that the performance was better than those of other methods, such as artificial neural networks (ANNs), support vector machines (SVMs), random forests (RFs), and deep LSTM. In order to improve the accuracy of HAR systems, more complex deep learning models have been proposed. Tan et al. used smartphone sensors for HAR. They proposed an ensemble learning algorithm (ELA) that combined a gated recurrent unit (GRU), a hybrid CNN+GRU, and a multilayer neural network, then fused them with the fully connected three layers [36]. In 2020, the International Data Corporation (IDC) reported that wearable devices are being used more frequently to monitor health due to the COVID-19 pandemic, resulting in a 35.1% increase in smartwatch sales [37]. Thus, more activities could be classified and higher accuracies could be approached if smartphones and smartwatches are synchronously used for HAR. Weiss et al. used smartphone and smartwatch sensors for HAR with an RF algorithm [38]. Mekruksavanich et al. also used smartphone and smartwatch sensors for HAR with a hybrid deep learning model called CNN+LSTM [39]. Prior studies have shown that adding hand-movement signals measured by smartwatch sensors can enhance the accuracy of HAR.

To improve the accuracy of HAR systems, the development of more complex deep learning models will be necessary. Thus, this study focuses on recognizing 18 different physical activities, including body and hand movements, as well as eating movements, utilizing data from sensors embedded in smartphones and smartwatches. The recognition process involves two steps: feature extraction and HAR. To extract features, a hybrid structure was used that consisted of a CNN and a recurrent neural network (RNN), while a multilayer perceptron neural network (MPNN) was used for the recognition of activities. The RNN was replaced with various other models, such as LSTM, GRU, BiLSTM, and bidirectional GRU, to optimize the hybrid structure. The MPNN was trained separately using backpropagation (BP), the ELM, and the regularized ELM (RELM). The HAR dataset used in this study was obtained from the UCI Machine Learning Repository and specifically the WISDM smartphone and smartwatch activity and biometrics dataset [31]. According to the experimental results, the proposed HAR system demonstrated superior performance when compared to the systems developed in existing studies.

## 2. Materials and Methods

The proposed HAR system has three components: a data processing unit, a feature extraction unit, and a classification unit, as illustrated in Figure 1. Physical activity signals are captured by a smartphone and a smartwatch and are subsequently sampled, segmented, and reshaped for further processing. The sensor data features are extracted using a hybrid CNN+RNN model. Finally, an MPNN is employed to classify the 18 types of physical activities.

### 2.1. UCI-WIDSM Dataset

The UCI-WISDM dataset [40] is comprised of tri-axial accelerometer and gyroscope data obtained from 51 volunteer subjects. The subjects carried an Android phone (a Google Nexus 5/5x or a Samsung Galaxy S5) in a front pocket of their pants and wore an Android watch (an LG G Watch) on their wrist while performing eighteen activities, which were categorized as body movements (walking, jogging, walking up stairs, sitting, and standing) included in many previous studies, hand movements (kicking, dribbling, catching, typing, writing, clapping, brushing teeth, and folding clothes) representing activities of daily life, and eating movements (eating pasta, drinking soup, eating a sandwich, eating chips, and drinking from a cup) to investigate the feasibility of automatic food-tracking applications [38]. The data were sampled at a rate of 20 Hz, and the 12 signals were segmented into fixed-width sliding windows of 6.4 s with 50% overlap between them. Each sample contained 12-channel signals, and each channel comprised 128 points. Samples containing two activities were removed. The numbers of training and testing samples were 34,316 and 14,707, respectively, and the sample numbers for each of the eighteen activities are presented in Table 1.

### 2.2. Feature-Extraction Model

Figure 2 illustrates a feature-extraction model that employs a hybrid CNN and RNN to extract the features of sensor signals. The fully connected layer, consisting of three layers, is used to classify the 18 types of physical activities. After training, the outputs of the RNN for the training samples serve as the feature samples to train the activation-classification models. Since the human movements in each activity occur in chronological order, the sensor signals represent time-sequential data. To address this, a time-distributed layer comprising four 1D CNNs (i.e., four pairs of CNNs with three layers and a maximal pool layer as the last layer) is stacked on top of the RNN. This separates a sample into four segments, with each segment containing 32 points. In the convolutional layer, the number of filters is 64; the kernel sizes are 3, 5, and 13; the stride is 1; and the padding is 4. In the pooling layer, the kernel size is 2, and the stride is 2. The activation function employed is ReLU. The RNN is replaced with the LSTM, BiLSTM, GRU, or BiGRU, with the unit numbers of LSTM and GRU set to 128 and those of BiLSTM and BiGRU set to 256. The batch size is set to 32, with the control reset gate and update gate using a sigmoid function and the hidden state using a tanh function. The numbers of full connection layers are 128, 64, and 18, respectively, with ReLU used as the activation function in hidden layers and softmax in the output layer. The loss function is the categorical Cross-Entropy (CE) function, and the Adam optimizer is used [41], with the learning rate set to 0.0001. Equation (1) is the formula for categorical CE:(1)$$\mathrm{CE}=-\log\left(\frac{\exp\left(a_{k}\right)}{\sum_{i=1}^{M}\exp\left(a_{i}\right)}\right)$$
where *M* is 18, *a_k_* is the score of softmax for the positive class, and *a_i_* is the score inferred by the net for each class.

### 2.3. Activation-Classification Model

The activation-classification model is a single-layer feedforward neural network (SLFN) with the ELM algorithm [42]. Its advantages are the convergent time being shorter than that of the BP method and its not converging to the local minimum. For an SLFN, a training set S = {(*X_r_*, *Y_ir_*| *X_i_* = (*x_r_*_1_, *x_r_*_2_, …, *x_rn_*)*^T^* ∈ *R_n_*, *Y_r_* = (*y_r_*_1_, *y_r_*_2_, …, *y_rm_*)^T^ ∈ *R_m_*}, where *X_r_* denotes the *r*th input vector and *Y_r_* represents the *r*th target vector. The output *o* of SLFN with *l* hidden neurons can be expressed as: (2)$$o_{k}=\overset{l}{\underset{j=1}{\sum }}\beta_{kj}f\left(W_{ij}X_{ij}+b_{j}\right),\;k=1,\ldots ,\;m,$$
where *f*(*x*) is the activation function in the hidden layer, *W_ji_* is the weight vector from the input layer to the *j*th hidden node, *W_ji_* = (*w_j_*_1_, *w_j_*_2_, …,*w_jn_*) ∈ R*_n_*, *b_j_* is the bias of the *j*th hidden node, *β_k_* is the weight vector from the hidden nodes to *k*th output layer, and *l* is the number of hidden layers. In the ELM, activation functions are nonlinear functions that provide nonlinear mapping for the system. *O_r_* is the *r*th output vector. Mean square error (MSE) is the object function:(3)$$\mathrm{MSE}=\overset{N}{\underset{i=1}{\sum }}{\left(Y_{i}-O_{i}\right)}^{2},$$
where *N* is the number of samples. The MSE will approach 0 as the number of hidden nodes approaches to infinity. The output *o* of SLFN is equal to the target output *y*. Thus, Equation (2) could be described as follows:(4)$$y_{k}=\overset{l}{\underset{j=1}{\sum }}\beta_{kj}f\left(W_{ij}X_{ij}+b_{j}\right),\;k=1,\ldots ,\;m.$$
*Y = Hβ*,
(5)
where *Y* is the output matrix, *H* is the matrix of the activation function in the hidden layer, and *β* is the weight matrix from the hidden nodes to the output layer. ELM uses random parameters *W_ij_* and *b_j_* in its hidden layer, and they are frozen during the whole training process.
*β = H*^†^*Y*,
(6)
where *H*^†^ is the Moore–Penrose inverse. The resident, *ε**_i_*, is between the target and output values of the *i*th sample.

However, the ELM has the risk to approach the result of over-fitting model because it bases on the empirical risk minimization principle [43]. Den et al. proposed a regularized ELM (RELM) that used a weight factor γ for empirical risk [44].
(7)$$\min\frac{1}{2}{\left\|\beta \right\|}^{2}+\frac{1}{2}\gamma {\left\|\varepsilon \right\|}^{2},$$
In order to obtain a robust estimate weakening outlier interference, *ε_i_* can be weighted by a factor *v_i_*. Equation (7) is changed thus:(8)$$\min\frac{1}{2}{\left\|\beta \right\|}^{2}+\frac{1}{2}\gamma {\left\|D\varepsilon \right\|}^{2}$$
where $D=dia\log\left(v_{1},\;v_{2},\;\ldots ,\;v_{N}\right)\;$ and $\varepsilon =\left[\varepsilon_{1},\;\varepsilon_{2},\ldots ,\varepsilon_{N}\right]$. The method of Lagrange multipliers is used to search for the optimal solution of Equation (8):(9)$$L\left(\beta ,\;\varepsilon ,\;\alpha \right)=\frac{1}{2}{\left\|\beta \right\|}^{2}+\frac{\gamma }{2}{\left\|D\varepsilon \right\|}^{2}-\alpha \left(H\beta -O-\varepsilon \right)$$
where *α* is the Lagrange multiplier with the equality constraints of Equation (9). Setting the gradients of *L*(*β*,*ε*,*α*) equal to zero gives the following Karush–Kuhn–Tucker (KKT) optimality conditions [44,45]:(10)$$\alpha =-\gamma {\left(H\beta -T\right)}^{T}$$
(11)$$\beta ={\left(\frac{I}{\gamma }+H^{T}D^{2}H\right)}^{†}H^{T}D^{2}T$$
(12)$$\varepsilon_{i}=\frac{\alpha_{i}}{\gamma },\;\left(i=1,\;2,\;\ldots ,N\right)$$

### 2.4. Experimental Protocol

The hardware used in this study comprised an Intel Core i7-8700 CPU and a GeForce GTX1080 GPU. The operating system used was Ubuntu 16.04LTS, with development being conducted in Anaconda 3 for Python 3.7. The deep learning tool used was Pytorch 1.10, and the compiler used was Jupyter Notebook. To assess the proposed method’s performance, we evaluated the optimal feature-extraction model and the activation-classification model for HAR separately.

In the feature-extraction model, the RNN was replaced with LSTM, BiLSTM, GRU, and BiGRU, separately. The training samples were used to adjust the parameters of the hybrid CNN+RNN, while the testing samples were used to evaluate the performances of these RNNs. The feature-extraction model that achieved the best performance was one in which the RNN outputs for all training and testing samples were used as the new training and testing samples to evaluate the activation-classification model.

In the activation-classification model, a multilayer perceptron neural network (MPNN) was used to classify the 18 physical activities. The output number of the MPNN was 18, and the input number depended on the number of RNN outputs. The training algorithms used were BP, ELM, and RELM. The number (*l*) of hidden layers and the regularized parameter (*γ*) of RELM were optimized using the grid-search method to find the optimal values.

### 2.5. Statistical Analysis

According to the proposed method, a sample was considered a true positive (TP) when the classification activity was correctly recognized, as a false positive (FP) when the classification activity was incorrectly recognized, as a true negative (TN) when the activity classification was correctly rejected, and as a false negative (FN) when the activity classification was incorrectly rejected. In this work, the performance of the proposed method was evaluated using the measures given by Equations (13)–(16):(13)$$Precision\;\left(\%\right)=\frac{\mathrm{TP}}{\mathrm{TP}+\mathrm{FP}}\times 100\%$$
(14)$$Recall\;\left(\%\right)=\frac{\mathrm{TP}}{\mathrm{TP}+\mathrm{FN}}\times 100\%$$
(15)$$F_{1}-score\;\left(\%\right)=\frac{2\times precision\times Recall}{Precision+Reacll}\times 100\%$$
(16)$$Accuracy\;\left(\%\right)=\frac{\mathrm{TP}+\mathrm{TN}}{\mathrm{TP}\mp \mathrm{TN}+\mathrm{FP}+\mathrm{FN}}\times 100\%$$

## 3. Results

In order to evaluate the effectiveness of the proposed method, we will present three sets of results: those for the feature-extraction model, the activation-classification model, and the training times of the models.

### 3.1. Analysis of the Feature-Extraction Model

The learning curves for the hybrid CNN+LSTM model are depicted in Figure 3, where (a) and (b) represent the accuracy and loss curves, respectively. The blue line corresponds to the training data, while the original line corresponds to the validation data. The optimal values for the accuracy and loss function are achieved at epoch 29. When applied to the testing data, the model achieved an average precision, recall, *F*_1_*-score*, and accuracy of 93.8%, 93.8%, 93.8%, and 94.1%, respectively. The total training time for the model was 130.26 s. In Figure 4, the learning curves for the hybrid CNN+GRU model are presented, where (a) and (b) denote the accuracy and loss curves, respectively. The blue line represents the training data, while the original line represents the validation data. The optimal values for the accuracy and loss function are attained at epoch 28. When evaluated on the testing data, the model achieved an average precision, recall, *F*_1_*-score*, and accuracy of 92.6%, 92.6%, 92.5%, and 92.2%, respectively. The total training time for the model was 98.67 s. The learning curves for the hybrid CNN+BiLSTM structure are displayed in Figure 5, where (a) and (b) represent the accuracy and loss curves, respectively. The blue line corresponds to the training data, while the original line corresponds to the validation data. The optimal values for the accuracy and loss function are achieved at epoch 30. When applied to the testing data, the model achieved an average precision, recall, *F*_1_*-score*, and accuracy of 95.3%, 95.3%, 95.3%, and 95.3%, respectively. The total training time for the model was 138.86 s. In Figure 6, the learning curves for the hybrid CNN+BiGRU model are presented, where (a) and (b) denote the accuracy and loss curves, respectively. The blue line represents the training data, while the original line represents the validation data. The optimal values for the accuracy and loss function are attained at epoch 29. When evaluated on the testing data, the model achieved an average precision, recall, *F*_1_*-score*, and accuracy of 95.7%, 95.4%, 95.5%, and 95.2%, respectively. The total training time for the model was 108.69 s. Table 2 provides an overview of the performances of four feature-extraction models. Although the hybrid structures with BiLSTM and BiGRU require more training time per epoch than LSTM and GRU (4.60 s vs. 4.49 s and 3.74 s vs. 3.52 s, respectively), their testing accuracies are superior to those of LSTM and GRU (95.3% vs. 94.1% and 95.2% vs. 92.2%). Given that the hybrid structure with BiGRU saves 19% of training time compared to BiLSTM and that their accuracies are very similar (95.25% vs. 95.3%), the feature-extraction model based on the hybrid CNN+BiGRU structure was chosen for building the HAR system.

### 3.2. Analysis of the Activation-Classification Model

To classify the 18 types of physical activities, an MPNN was utilized, where the input and output nodes were set to 256 and 18, respectively. The MPNN was trained using three activation-classification algorithms: BP, ELM, and RELM. The performance of ELM and RELM was influenced by two parameters: the regularized index (*γ*) and the number of hidden layers (*l*).

#### 3.2.1. Performance of the MPNN with the BP Algorithm

The MPNN with the BP algorithm had two hidden layers with 128 and 64 nodes, respectively, where ReLU was used as the activation function in the hidden layers and softmax in the output layer. Table 3 shows the performances of the MPNN with the BP algorithm for 18 physical activities on the testing data. The model achieved an average precision of 97.1%, an average recall of 97.2%, an average *F*_1_*-score* of 97.2%, and an accuracy of 97.2%. The total training time was 10.563 s. Among the 18 activities, the worst *F*_1_*-scores* were obtained for the eating pasta, catching a ball, and eating a sandwich activities, which all involve hand and eating movements.

#### 3.2.2. The Optimal Parameters of the RELM

The SLFN utilized both ELM and RELM algorithms, and the optimal parameters for the RELM were determined using a grid-search method. For the RELM, the regularized index (*γ*) was set to 5 × 10^−4^, and the number of hidden layers was gradually increased from 256 nodes to 8000 nodes. Table 4 displays the testing accuracies and training times for various numbers of hidden layers. The highest accuracy of 98.35% and a training time of 3.80 s were achieved with 6000 hidden nodes. After that, when *l* was fixed at 6000, *γ* gradually increased from 5 × 10^−4^ to 4. Table 5 shows the testing accuracies and training times for different regularized indexes. It was observed that the most accurate results and the highest training time were obtained when *γ* was set to 5 × 10^−4^. In Equation (7), the empirical risk, ${\left\|\varepsilon \right\|}^{2}$, is regularized by *γ*. Thus, the performances of the ELM and RELM would be close in this study. 

#### 3.2.3. Performances of the SLFN with the ELM and RELM Algorithms

For the ELM algorithm, the SLFN had one hidden layer with 6000 nodes. Figure 7 shows the confusion matrix of the classification of eighteen activities. The performances of writing, clapping, brushing teeth, eating chips, and drinking from a cup activities were better than those for the ELM algorithm. Table 6 presents the performances of the SLFN with the ELM algorithm on the testing data. The model achieved an average precision of 97.9%, a recall of 97.9%, an *F*_1_*-score* of 97.9%, and an accuracy of 97.8%. The total training time was 7.52 s. The *F*_1_*-scores* for the eating pasta, catching a ball, and eating a sandwich activities rose to 98.0%, 96.4%, and 98.1%, respectively.

For the RELM algorithm, *l* was set to 6000 for the SLFN, and *γ* was set to 5 × 10^−4^. Figure 8 shows the confusion matrix of the classification of eighteen activities. The eating pasta activity was easily confused with the drinking soup and drink from a cup activities. Catching a ball was easily confused with kicking a ball. Table 7 shows the performances of the SLFN with the RELM algorithm on the testing data. The model achieved an average precision of 98.3%, a recall of 98.4%, an *F*_1_*-score* of 98.4%, and an accuracy of 98.3%. The total training time was 3.59 s. The *F*_1_*-scores* for the eating pasta, catching a ball, and eating a sandwich activities rose to 98.1%, 97.6%, and 99.2%, respectively.

## 4. Discussion

The proposed HAR system involves the use of a hybrid CNN+RNN model to extract activation features from accelerometers and gyroscopes in smartphones and smartwatches. This method was originally proposed by Tan et al. [36]. Since the accelerometer and gyroscope signals for activities are time-sequential, the performance of different RNN models can vary for HAR. In this study, LSTM, GRU, BiLSTM, and BiGRU were explored, and the classifying performances of BiLSTM and BiGRU were found to be very similar. However, BiGRU had a shorter training time than BiLSTM (108.69 s vs. 138.86 s) and was therefore used to extract the activation features. To enhance the performance of the classifier, the SLFN with the RELM algorithm was used. The ELM algorithm, which utilizes an SLFN with hidden neural weights and bias, was proposed by Huang et al. [46,47]. The ELM has an extremely fast training time and good generalized performance. Deng et al. proposed the RELM, which is based on the structural risk minimization principle of statistical learning theory and overcomes the drawbacks of the ELM [44]. Table 8 summarizes the total performances of the activation-classification models, the MLNN with BP, and the SLFN with the ELM and RELM. It was found that the classifying performances of the ELM and RELM were very similar (97.8% vs. 98.2% accuracies). The reason for this was the very small regularized weight, γ. However, the training time of the ELM was shorter (7.52 s vs. 10.56 s). However, the RELM exhibited the best performance for HAR despite its longer total testing time (feature extraction plus classification) compared to the ELM (0.038 s vs. 0.025 s).

Table 9 presents a comparative analysis of our proposed method with those of other studies that utilized the UCI-WISDM smartphone and/or smartwatch activity and biometrics dataset for six/eighteen activities. Previous studies [36,48,49,50,51,52] only classified six activities, while studies [38,39] classified eighteen activities. As shown, the proposed HAR system using the hybrid CNN+BiGRU model and the SLFN with the RELM achieved an *F*_1_*-score* and an accuracy of 98.4% and 98.2%, respectively, which are among the best results reported in the literature.

For the opening HAR datasets, the sensors, which are all accelerometers and gyroscopes, are embedded in smartphones or smartwatches or are body-worn [38,52]. The greater the number of sensors, the higher the accuracy of HAR. Table 10 displays the *F*_1_*-scores* of 18 physical activities using the accelerometers and gyroscopes embedded in the smartphones and smartwatches. We explored the performance of our proposed method when only using these sensors, specifically, either the accelerometers or the gyroscopes. When HAR used the sensors of the smartphones and smartwatches, the average *F*_1_*-scores* were 90.7% and 89.1%, respectively. When only the accelerometers or gyroscopes of the smartphones and smartwatches were used for HAR, the average *F*_1_*-scores* were 94.1% and 76.9%, respectively. These results suggest that the accelerometers provide more information than the gyroscopes for HAR.

## 5. Conclusions

The proposed deep learning model utilizes the hybrid CNN+BiGRU for feature extraction from the signals of sensors embedded in smartphones and smartwatches and the SLFN with the RELM algorithm for the classification of 18 physical activities, including body, hand, and eating movements. The experimental results demonstrate that the proposed model outperforms other existing schemes that utilize deep learning or machine learning methods in terms of *F*_1_*-scores* and accuracy. Notably, the worst *F*_1_*-score* was found in the classification for brushing teeth. Our investigation shows that using different deep learning models for feature extraction and classification during the training phase can effectively increase recognition accuracy and training time. Moreover, since the data are recorded by smartphones and smartwatches, our proposed method has the potential to be used for mHealth in real time in environments without embedding of wireless sensor networks. The weakness of this study is that it ignores signals sent when two activities are transferring. Thus, in the future, we will explore this problem.

## Figures and Tables

**Figure 1 sensors-23-03354-f001:**
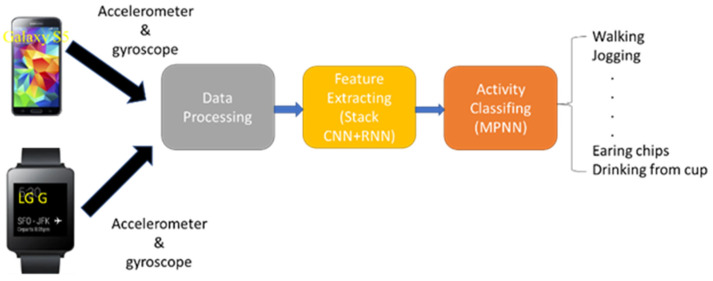
Structural diagram of the proposed HAR system, including the data processing unit, the feature extraction unit, and the classification unit.

**Figure 2 sensors-23-03354-f002:**
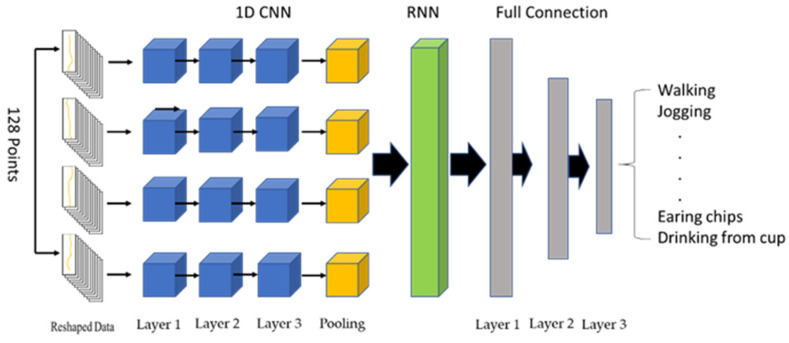
Structural diagram of the feature-extraction model.

**Figure 3 sensors-23-03354-f003:**
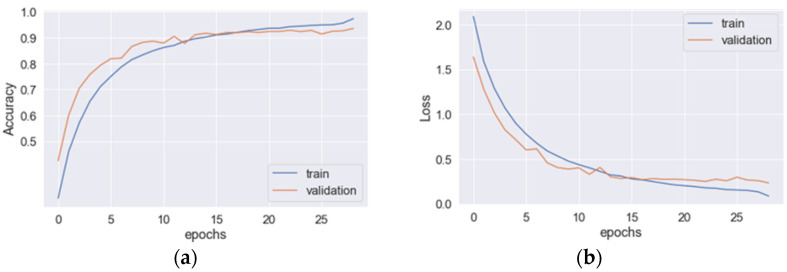
The learning curves for the hybrid CNN+LSTM model: (**a**) the accuracy and (**b**) loss curves.

**Figure 4 sensors-23-03354-f004:**
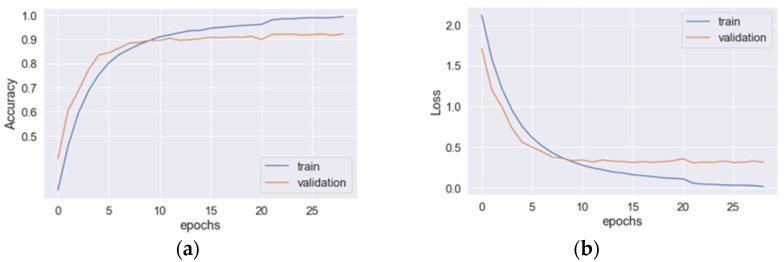
The learning curves for the hybrid CNN+GRU model: (**a**) the accuracy and (**b**) loss curves.

**Figure 5 sensors-23-03354-f005:**
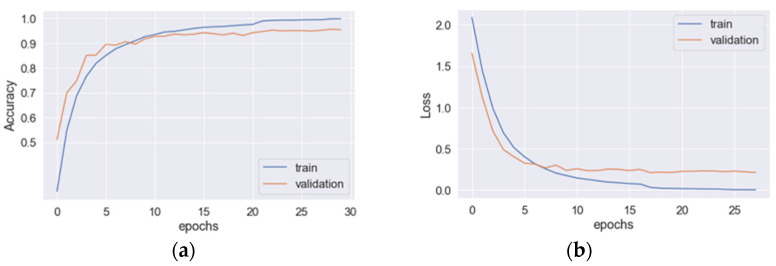
The learning curves for the hybrid CNN+BiLSTM model: (**a**) the accuracy and (**b**) loss curves.

**Figure 6 sensors-23-03354-f006:**
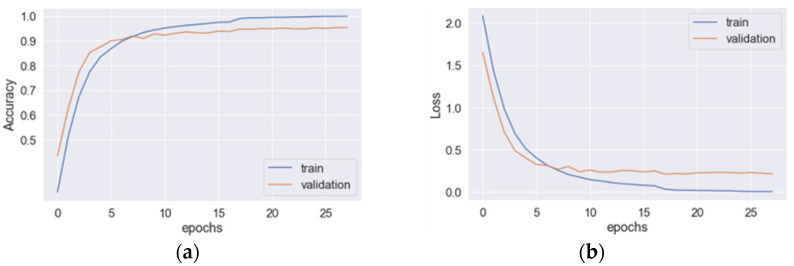
The learning curves for the hybrid CNN+BiGRU model: (**a**) the accuracy and (**b**) loss curves.

**Figure 7 sensors-23-03354-f007:**
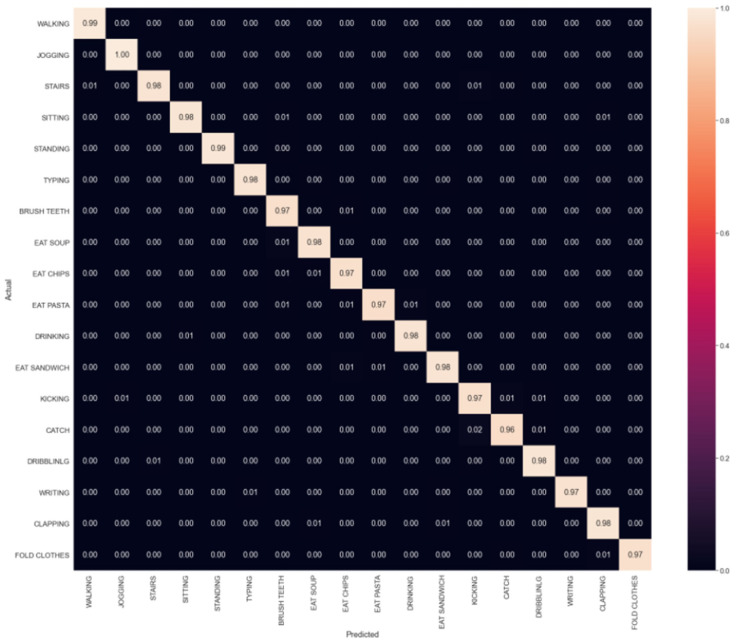
The confusion matrix of the classification of eighteen activities for the ELM algorithm.

**Figure 8 sensors-23-03354-f008:**
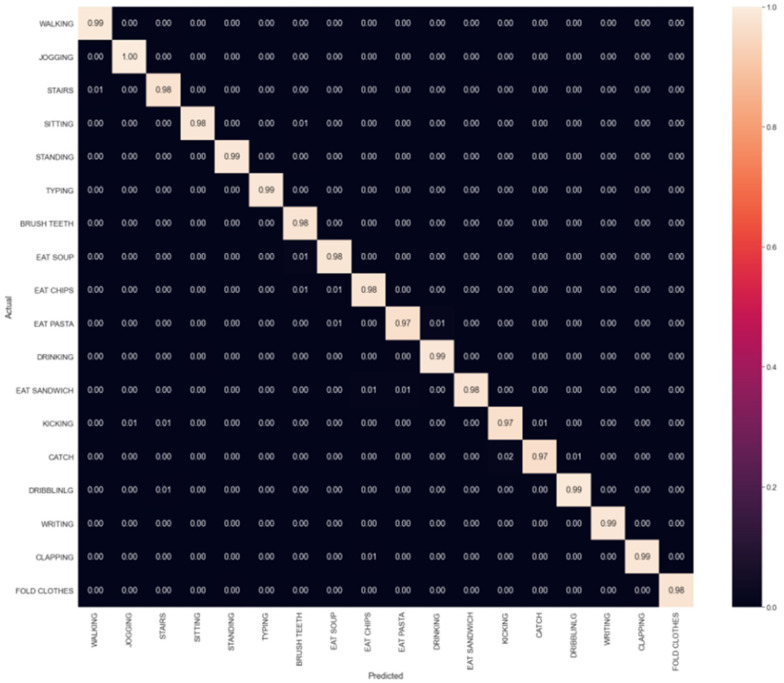
The confusion matrix for the classification of eighteen activities for the RELM algorithm.

**Table 1 sensors-23-03354-t001:** Sample numbers of eighteen activities for model training and testing with the UCI-WISDM dataset.

Activity	Training Number	Testing Number
Walking	1921	807
Jogging	1901	827
Walking up stairs	1920	808
Sitting	1895	833
Standing	1891	837
Kicking (soccer ball)	1932	797
Dribbling (basketball)	1906	822
Catching (tennis ball)	1893	835
Typing	1885	843
Writing	1880	766
Clapping	1945	783
Brushing teeth	1876	852
Folding clothes	1919	809
Eating pasta	1915	814
Drinking soup	1928	800
Eating a sandwich	1950	778
Eating chips	1898	830
Drinking from a cup	1861	866

**Table 2 sensors-23-03354-t002:** The performances of the feature-extraction models with LSTM, GRU, BiLSTM, and BiGRU, separately.

RNN	Precision (%)	Recall (%)	*F*_1_*-Score* (%)	Accuracy (%)	Training Time (s/epoch)
LSTM	93.8	93.8	93.1	94.1	4.49
GRU	92.6	92.6	92.5	92.2	3.52
BiLSTM	95.3	95.3	95.3	95.3	4.60
BiGRU	95.7	95.4	95.5	95.2	3.74

**Table 3 sensors-23-03354-t003:** The performances of the MPNN with the BP algorithm for 18 types of physical activities.

	Precision (%)	Recall (%)	*F*_1_*-Score* (%)	Accuracy (%)
Walking	97.2	98.0	97.6	97.2
Jogging	97.3	98.8	98.0	
Stairs	97.7	97.0	97.3	
Sitting	98.0	97.2	97.6	
Standing	98.6	98.0	98.3	
Kicking	95.8	96.4	96.1	
Dribbling	96.6	97.1	96.8	
**Catching a ball**	95.9	95.0	**95.4**	
Typing	98.8	99.1	98.9	
Writing	99.0	98.5	98.7	
Clapping	97.5	98.0	97.7	
Brushing teeth	97.3	97.3	97.3	
Folding clothes	98.0	99.1	98.5	
**Eating pasta**	95.0	94.9	**94.9**	
Drinking soup	96.6	96.6	96.6	
**Eating a sandwich**	95.1	96.2	**95.6**	
Eating chips	96.8	96.9	96.8	
Drinking from a cup	96.7	96.3	96.5	
Average	97.1	97.2	97.2	

**Table 4 sensors-23-03354-t004:** The testing accuracies and training times for various numbers of hidden layers with *γ* set at 5 × 10^−4^.

N	Accuracy (%)	Training Time (s)	N	Accuracy (%)	Training Time (s)
256	97.10%	2.49	2000	97.85%	2.654
300	97.33%	2.402	2500	97.88%	2.757
400	97.54%	2.414	3000	97.91%	2.892
500	97.60%	2.393	3500	97.97%	2.989
600	97.60%	2.444	4000	97.98%	3.093
700	97.65%	2.492	4500	98.01%	3.223
800	97.70%	2.528	5000	98.05%	3.317
900	97.74%	2.506	5500	98.15%	3.466
1000	97.76%	2.603	**6000**	**98.25%**	**3.802**
1100	97.78%	2.592	6500	98.02%	3.826
1200	97.81%	2.600	7000	98.05%	3.886
1300	97.81%	2.617	7500	97.99%	3.894
1400	97.82%	2.622	8000	98.05%	4.116
1500	97.83%	2.624			

**Table 5 sensors-23-03354-t005:** The testing accuracies and training times for the different regularized indexes with *l* set at 6000.

γ	Accuracy (%)	Training Time (s)
4	50.69	3.530
2	96.95	3.348
1	97.70	3.359
5 × 10^−1^	97.80	3.414
1 × 10^−1^	97.82	3.616
5 × 10^−2^	97.85	3.484
1 × 10^−2^	97.86	3.512
5 × 10^−3^	97.92	3.607
1 × 10^−3^	98.04	3.584
**5 × 10^−4^**	**98.25**	**3.802**

**Table 6 sensors-23-03354-t006:** The performances of SLFN with the ELM algorithm for 18 types of physical activities.

	Precision (%)	Recall (%)	*F*_1_-*Score* (%)	Accuracy (%)
Walking	99.1	99.3	99.2	97.8
Jogging	100.0	100.0	100.0	
Stairs	98.2	97.9	98.0	
Sitting	97.8	98.2	98.0	
Standing	98.5	99.1	98.8	
Kicking	95.2	96.8	96.0	
Dribbling	98.3	97.8	98.0	
**Catching a ball**	96.6	96.2	**96.4**	
Typing	98.0	97.8	97.9	
Writing	99.1	97.0	98.0	
Clapping	97.2	98.1	97.6	
Brushing teeth	95.4	97.1	96.2	
Folding clothes	99.0	97.5	98.2	
**Eating pasta**	98.3	97.7	**98.0**	
Drinking soup	97.6	98.5	98.0	
**Eating a sandwich**	97.7	98.5	**98.1**	
Eating chips	96.9	96.9	96.9	
Drinking from a cup	98.7	98.0	98.3	
Average	97.9	97.9	97.9	

**Table 7 sensors-23-03354-t007:** The performances of the SLFN with the RELM algorithm for 18 types of physical activities.

	Precision (%)	Recall (%)	*F*_1_-*Score* (%)	Accuracy (%)
Walking	99.1	99.2	99.1	98.25
Jogging	99.4	100.0	99.7	
Stairs	97.8	97.8	97.8	
Sitting	98.2	98.3	98.2	
Standing	99.0	99.7	99.3	
Kicking	97.3	97.0	97.1	
Dribbling	98.1	98.7	98.4	
**Catching a ball**	97.9	97.3	97.6	
Typing	99.1	99.0	99.0	
Writing	98.8	98.8	98.8	
Clapping	98.0	97.0	97.5	
Brushing teeth	96.3	97.6	96.9	
Folding clothes	99.1	98.8	98.9	
**Eating pasta**	98.0	98.3	98.1	
Drinking soup	97.7	97.9	97.8	
**Eating a sandwich**	100.0	98.4	99.2	
Eating chips	97.9	98.0	97.9	
Drinking from a cup	98.5	99.1	98.8	
Average	98.3	98.4	98.4	

**Table 8 sensors-23-03354-t008:** Total performances of activation-classification models: MLNN with BP and SLFN with the ELM and RELM.

	MPNN with BP	SLFN with ELM	SLFN with RELM
Precision (%)	97.1	97.9	98.3
Recall (%)	97.2	97.9	98.4
*F*_1_-*score* (%)	97.2	97.9	98.4
Accuracy (%)	97.2	97.8	98.2
Training time (s)	10.56	7.52	3.59
Total testing time (s)	0.103	0.025	0.038

**Table 9 sensors-23-03354-t009:** Comparative results of various methods using the UCI-WISDM dataset.

Ref.	Classification Method	Activities/ Wearable Devices	*F*_1_-*Score* (%)	Accuracy (%)
[36]	CNN+GRU	6/phone	91.7	NA
[38]	Riege forest	18/phone and watch	NA	94.4
[39]	CNN+LSTM	18/watch	96.3	96.2
[48]	CNN+handcrafted features	6/phone	NA	93.3
[49]	ConvAS	6/phone	NA	94.9
[50]	CNN+LSTM	6/phone and watch	NA	96.0
[51]	Hesitant fuzzy belief structures	6/phone and watch	NA	95.82
[52]	ConvAE-LSTM	6/Phone	97.4	97.1
Proposed method	Hybrid CNN+BGRU SLFN with RELM	18/phone and watch	98.4	98.2

**Table 10 sensors-23-03354-t010:** *F*_1_*-scores* of 18 physical activities using the accelerometers and gyroscopes embedded in the smartphones and smartwatches.

Activities	Phone		Watch		Phone	Watch	Acce.	Gyro.	All
	Acce.	Gyro.	Acce.	Gyro.					
Walking	96.3	93.1	94.8	89.7	99.2	96.3	98.1	83.2	99.2
Jogging	97.0	97.1	98.5	94.1	97.8	98.7	97.6	96.7	99.7
Stairs	88.2	79.7	80.0	69.7	92.3	89.2	95.12	78.7	97.8
Sitting	83.6	40.5	80.6	55.0	91.4	87.5	94.3	68.4	98.3
Standing	88.3	58.1	89.2	61.8	93.7	90.7	93.1	68.8	99.3
Kicking	79.8	70.4	87.7	77.7	90.0	89.7	92.9	77.0	97.2
Dribbling	84.4	60.5	91.6	74.14	90.5	93.6	95.2	87.3	98.4
Catching	76.2	70.5	95.3	80.4	87.2	91.7	89.0	90.3	97.6
Typing	91.3	40.0	94.3	77.2	90.6	96.0	94.2	81.2	99.1
Writing	89.6	54.4	88.7	71.8	91.8	89.9	96.2	78.7	98.8
Clapping	89.0	77.4	96.0	83.3	91.7	96.5	94.3	94.4	97.5
Brushing teeth	87.6	61.7	95.5	75.5	88.39	97.0	92.5	89.9	97.0
Folding clothes	82.5	62.2	90.7	67.2	91.1	95.1	95.0	83.0	98.9
Eating pasta	85.5	25.5	77.3	54.5	89.6	85.4	90.9	63.1	98.2
Drinking soup	80.1	27.6	78.5	56.2	84.7	83.1	93.1	66.5	97.8
Eating a sandwich	83.6	15.6	48.7	26.3	90.6	69.8	94.8	42.2	99.2
Eating chips	81.4	21.0	66.8	41.4	81.9	71.7	89.3	50.0	98.0
Drinking from a cup	85.7	26.4	77.7	55.8	89.7	81.9	96.2	67.1	98.8
Average	86.2	54.6	85.2	67.4	90.7	89.1	94.1	76.0	98.4

## Data Availability

The data are available from https://archive.ics.uci.edu/ml/datasets/WISDM+Smartphone+and+Smartwatch+Activity+and+Biometrics+Dataset+.

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
