# Peer review of "Using a Hybrid Neural Network and a Regularized Extreme Learning Machine for Human Activity Recognition with Smartphone and Smartwatch"

_sensors, 2023, doi:10.3390/s23063354_

Round 1

Reviewer 1 Report

This is a carefully done study and the findings are of considerable interest. A few minor revisions are list below.

(1) In the preface, the author should briefly introduce and point out the advantages and disadvantages of the current research algorithm, and What is the problem the paper needs to be solved.

(2)  This paper lacks specific explanations and analysis of experimental results, for example, this paper does not explain the reasons why the larger the γ value, the worse the effect.

(3) The result graphs in 3.1 should be put together to facilitate visual comparison.

(4) In 2.4, when comparing the feature-extraction mode part, the activation-classification mode part is not explained, and vice versa.

(5)  In the part of evaluation strategy in Experiments in 3.2.3 : the author should add confusion matrix to make it easier to see the categories of misidentification, and the results should be interpreted. 

(6) When conducting comparative experiments, the test data sets were not presented. There should be several examples of extracted data information for the reader to understand.

Author Response

Reviewer 1 (round 1)

Dear Anonymous Reviewer,

The authors are grateful to your comments and suggestions for improving the quality and presentation of this paper. All comments are followed. The revised parts are highlighted in red. It is our sincere hope that this revision will enhance readability and strengthen of the manuscript to satisfy the requirements of this prestigious journal.

Comments and Suggestions for Authors

This is a carefully done study and the findings are of considerable interest. A few minor revisions are list below.

(1) In the preface, the author should briefly introduce and point out the advantages and disadvantages of the current research algorithm, and What is the problem the paper needs to be solved.

ANS: We modified two paragraphs in Introduction sector.

Line: 66-102

Convolutional neural networks (CNN) can extract features from signals, while long short-term memory (LSTM) can recognize time-sequential features. Therefore, some studies have proposed deep neural networks that combine CNN and LSTM to recognize physical activities [22, 23]. Li et al. utilized bidirectional LSTM (BiLSTM) for continuous human activity recognition (HAR) and fall detection with soft feature fusion between the signals measured by wearable sensors and radar [24]. The extreme learning machine (ELM) has shown excellent results in classification tasks with extremely fast learning speed [25]. Chen et al. proposed an ensemble ELM algorithm for HAR using smartphone sensors [26]. Their results showed that the performance was better than other methods, such as artificial neural network (ANN), support vector machine (SVM), random forest (RF), and deep LSTM. In order to improve the accuracy of HAR system, the more complex deep learning models have been proposed. Tan et al. used smartphone sensors for HAR by proposing an ensemble learning algorithm (ELA) that combined gated recurrent unit (GRU), hybrid CNN+GRU, and multilayer neural network, then fused them with the fully connected layer with three layers [27]. In 2020, the International Data Corporation (IDC) reported that wearable devices are being used more frequently to monitor health due to the COVID-19 pandemic, resulting in a 35.1% increase in smartwatch sales [30]. Thus, the more activities could be classified, and the higher accuracy could be approached when the smartphone and smartwatch were synchronously used for HAR. Weiss et al. used smartphone and smartwatch sensors for HAR with the RF algorithm [28]. Mekruksavanich et al. also used smartphone and smartwatch sensors for HAR with a hybrid deep learning model called CNN+LSTM [29]. Prior studies have shown that adding hand-movement signals measured by smartwatch sensors can enhance the accuracy of HAR.

To improve the accuracy of HAR systems, the developments of more complex deep learning models would be necessary. Thus, the aim of this study focuses on recognizing 18 different physical activities, including body and hand movements, as well as eating movements, by utilizing data from sensors embedded in the smartphone and smartwatche. The recognition process involves two parts: feature extraction and HAR. To extract features, a hybrid structure was used that consisted of a CNN and recurrent neural network (RNN), while a multilayer perceptron neural network (MPNN) was used for the recognition of activities. The RNN was replaced with various other models such as LSTM, GRU, BiLSTM, and bidirectional GRU to optimize the hybrid structure. The MPNN was trained separately using backpropagation (BP), ELM, and regularized ELM (RELM). The HAR dataset used in this study was obtained from the UCI Machine Learning Repository and specifically the WISDM smartphone and smartwatch activity and biometrics dataset [31]. According to the experimental results, the proposed HAR system demonstrated superior performance when compared to existing studies.

(2)  This paper lacks specific explanations and analysis of experimental results, for example, this paper does not explain the reasons why the larger the γ value, the worse the effect.

ANS: We modified the texts in 3.2.2 of Results and Discussion sectors to explain the affection of γ value in RELM. Moreover, we added the confusion matrices in 3.2.3 of Results sector to analyze which kinds of activities are easy to be misclassified.

Line: 287-297

The SLFN utilized both ELM and RELM algorithms, and the optimal parameters for RELM were determined using a grid-search method. For RELM, the regularized index (γ) was set to 5×10-4, and the number of hidden layers was gradually increased from 256 nodes to 8000 nodes. Table 3 displays the testing accuracies and training time for various numbers of hidden layers. The highest accuracy of 98.35% and a training time of 3.80 seconds were achieved with 6000 hidden nodes. After that, when l was fixed at 6000, the γ was gradually increased from 5×10-4 to 4. Table 4 shows the testing accuracies and training times for different regularized indexes. It was observed that the most accurate results and the highest training time were obtained when γ was set to 5×10-4. In Eq.(7), the empirical risk, , is regularized by γ. Thus, the performances of ELM and RELM would be close in this study.

Line: 303-310

For the ELM algorithm, the SLFN had one hidden layer with 6000 nodes. Figure 8 shows the confusion matrix of the classification of eighteen activities. The performances of writing, clapping, brushing teeth, eating chips, and drinking from cup activities are better than the ELM algorithm. Table 5 presents the performances of the SLFN with ELM algorithm on the testing data. The model achieved an average precision of 97.9%, recall of 97.9%, F1-score of 97.9%, and accuracy of 97.8%, respectively. The total training time was 7.52 seconds. The F1-scores for the eating pasta, catching ball, and eating sandwich activities arise to 98.0%, 96.4%, and 98.1%, respectively.

Line: 314-321

For the RELM algorithm, the SLFN had l with 6000, and γ was set to 5×10-4. Figure 8 shows the confusion matrix of the classification of eighteen activities. The eating past activity would be easily confused by the drinking soup and drink from cup activities. The catching ball would be easily confused by the kicking ball. Table 6 shows the performances of the SLFN with RELM algorithm on the testing data. The model achieved an average precision of 98.3%, recall of 98.4%, F1-score of 98.4%, and accuracy of 98.3%, respectively. The total training time was 3.59 seconds. The F1-scores for the eating pasta, catching ball, and eating sandwich activities arise to 98.1%, 97.6%, and 99.2%, respectively.

(3) The result graphs in 3.1 should be put together to facilitate visual comparison.

ANS: We modified typesetting of these paragraphs in 3.1 of Results sector.

Line: 216-247

The learning curves for the hybrid CNN+LSTM model are depicted in Figure 3, where (a) and (b) represent the accuracy and loss curves, respectively. The blue line corresponds to the training data, while the original line corresponds to the validation data. The optimal values for the accuracy and loss function are achieved at epoch 29. When applied to the testing data, the model achieved an average precision, recall, F1-score, and accuracy of 93.8%, 93.8%, 93.8%, and 94.1%, respectively. The total training time for the model was 130.26 seconds. In Figure 4, the learning curves for the hybrid CNN+GRU model are presented, where (a) and (b) denote the accuracy and loss curves, respectively. The blue line represents the training data, while the original line represents the validation data. The optimal values for the accuracy and loss function are attained at epoch 28. When evaluated on the testing data, the model achieved an average precision, recall, F1-score, and accuracy of 92.6%, 92.6%, 92.5%, and 92.2%, respectively. The total training time for the model was 98.67 seconds. The learning curves for the hybrid CNN+BiLSTM structure are displayed in Figure 5, where (a) and (b) represent the accuracy and loss curves, respectively. The blue line corresponds to the training data, while the original line corresponds to the validation data. The optimal values for the accuracy and loss function are achieved at epoch 30. When applied to the testing data, the model achieved an average precision, recall, F1-score, and accuracy of 95.3%, 95.3%, 95.3%, and 95.3%, respectively. The total training time for the model was 138.86 seconds. In Figure 6, the learning curves for the hybrid CNN+BiGRU model are presented, where (a) and (b) denote the accuracy and loss curves, respectively. The blue line represents the training data, while the original line represents the validation data. The optimal values for the accuracy and loss function are attained at epoch 29. When evaluated on the testing data, the model achieved an average precision, recall, F1-score, and accuracy of 95.7%, 95.4%, 95.5%, and 95.2%, respectively. The total training time for the model was 108.69 seconds. Table 1 provides an overview of the performances of four feature-extraction models. Although the hybrid structures with BiLSTM and BiGRU require more training time per epoch than LSTM and GRU (4.60 seconds vs 4.49 seconds and 3.74 seconds vs 3.52 seconds, respectively), their testing accuracies are superior to those of LSTM and GRU (95.3% vs 94.1% and 95.2% vs 92.2%). Given that the hybrid structure with BiGRU saves 19% of training time compared to BiLSTM, their accuracies are very similar (95.25% vs 95.3%), the feature-extraction model based on the hybrid CNN+BiGRU structure was chosen for building the HAR system.

(4) In 2.4, when comparing the feature-extraction mode part, the activation-classification mode part is not explained, and vice versa.

ANS: We have described the feature-extraction model in Figure 2 of 2.1 sector. The RNN were replaced with the LSTM, BiLSTM, GRU, or BiGRU. The feature-extraction model that achieved the best performance was one in which the RNN outputs for all training and testing samples were used as the new training and testing samples for evaluating the activation-classification model.

For the activation-classification model, it is a multilayer perceptron neural network (MPNN), which output number of the MPNN was 18, and the input number depended on the number of RNN. When using the BP algorithm, there were two hidden layers with 128 and 64 nodes. When using ELM and RELM algorithms, there was one hidden layer with 6000 nodes. These texts have been described in 2.4, 3.2.1, and 3.2.4 sectors.

Line: 192-203

In the feature-extraction model, the RNN was replaced with the LSTM, BiLSTM, GRU, and BiGRU, separately. The training samples were used to adjust the parameters of the hybrid CNN+RNN, while the testing samples were used to evaluate the performances of these RNNs. The feature-extraction model that achieved the best performance was one in which the RNN outputs for all training and testing samples were used as the new training and testing samples for evaluating the activation-classification model.

In the activation-classification model, a multilayer perceptron neural network (MPNN) was used to classify the 18 physical activities. The output number of the MPNN was 18, and the input number depended on the number of RNN. The training algorithms used were BP, ELM, and RELM. The number (l) of hidden layers and the regularized parameter (γ) of RELM were optimized using the grid-search method to find the optimal values.

Line: 276-284

3.2.1 Performances of MPNN with BP Algorithm

The MPNN with BP algorithm had two hidden layers with 128 and 64 nodes, respectively, where ReLU was used as the activation function in the hidden layers, and softmax in the output layer. Table 2 shows the performances of the MPNN with BP algorithm for 18 physical activities on the testing data. The model achieved an average precision of 97.1%, an average recall of 97.2%, an average F1-score of 97.2%, and an accuracy of 97.2%. The total training time was 10.563 seconds. Among the 18 activities, the worst F1-scores were obtained for the eating pasta, catching ball, and eating sandwich activities, which all involve hand and eating movements.

Line: 302-310

3.2.3 Performances of SLFN with ELM and RELM Algorithms

For the ELM algorithm, the SLFN had one hidden layer with 6000 nodes. Figure 8 shows the confusion matrix of the classification of eighteen activities. The performances of writing, clapping, brushing teeth, eating chips, and drinking from cup activities are better than the ELM algorithm. Table 5 presents the performances of the SLFN with ELM algorithm on the testing data. The model achieved an average precision of 97.9%, recall of 97.9%, F1-score of 97.9%, and accuracy of 97.8%, respectively. The total training time was 7.52 seconds. The F1-scores for the eating pasta, catching ball, and eating sandwich activities arise to 98.0%, 96.4%, and 98.1%, respectively.

(5)  In the part of evaluation strategy in Experiments in 3.2.3 : the author should add confusion matrix to make it easier to see the categories of misidentification, and the results should be interpreted.

ANS: We added the confusion matrices in the results of ELM and RELM algorithms.

Line: 302-312

For the ELM algorithm, the SLFN had one hidden layer with 6000 nodes. Figure 8 shows the confusion matrix of the classification of eighteen activities. The performances of writing, clapping, brushing teeth, eating chips, and drinking from cup activities are better than the ELM algorithm. Table 5 presents the performances of the SLFN with ELM algorithm on the testing data. The model achieved an average precision of 97.9%, recall of 97.9%, F1-score of 97.9%, and accuracy of 97.8%, respectively. The total training time was 7.52 seconds. The F1-scores for the eating pasta, catching ball, and eating sandwich activities arise to 98.0%, 96.4%, and 98.1%, respectively.

Figure 7. The confusion matrix of the classification of eighteen activities for the ELM algorithm.

Line: 314-323

For the RELM algorithm, the SLFN had l with 6000, and γ was set to 5×10-4. Figure 8 shows the confusion matrix of the classification of eighteen activities. The eating past activity would be easily confused by the drinking soup and drink from cup activities. The catching ball would be easily confused by the kicking ball. Table 6 shows the performances of the SLFN with RELM algorithm on the testing data. The model achieved an average precision of 98.3%, recall of 98.4%, F1-score of 98.4%, and accuracy of 98.3%, respectively. The total training time was 3.59 seconds. The F1-scores for the eating pasta, catching ball, and eating sandwich activities arise to 98.1%, 97.6%, and 99.2%, respectively.

Figure 8. The confusion matrix of the classification of eighteen activities for the RELM algorithm.

(6) When conducting comparative experiments, the test data sets were not presented. There should be several examples of extracted data information for the reader to understand.

ANS: We modified the texts in Table 8. We added the information about these previous studies classifying the number of activities and using wearable devices.

Line: 348-355

Table 8 presents a comparative analysis of our proposed method with other studies that utilized the UCI-WISDM Smartphone and / or Smartwatch Activity and Biometrics dataset for six / eighteen activities. Previous studies [27,37,38,39,40,41] only classified six activities, while studies [28,29] classified eighteen activities. As shown, the proposed HAR system using the hybrid CNN+BiGRU model and SLFN with RELM achieved an F1-score and accuracy of 98.4% and 98.2%, respectively, which is among the best results reported in the literature.

Table 8 Comparative result of various methods using UCI-WISDM dataset.

Ref.

Classification Method

Activities/

Wearable Devices

F1-Score (%)

Accuracy (%)

[27]

CNN+GRU

6 / Phone

91.7

NA

[28]

Riege Forest

18 / Phone & Watch

NA

94.4

[29]

CNN+LSTM

18 / Watch

96.3

96.2

[37]

CNN+ Handcrafted Features

6 / Phone

NA

93.3

[38]

ConvAS

6 / Phone

NA

94.9

[39]

CNN+LSTM

6 / Phone & Watch

NA

96.0

[40]

Hesitant Fuzzy Belief

Structures

6 / Phone & Watch

NA

95.82

[41]

ConvAE-LSTM

6 / Phone

97.4

97.1

Proposed method

Hybrid CNN+BGRU

SLFN with RELM

18 / Phone & Watch

98.4

98.2

Reviewer 2 Report

The paper proposes a HAR system for classifying 18 types of physical activities using data from sensors embedded in smartphones and smartwatches. The recognition 21 process consists of two parts: feature extraction and HAR.

The topic is interesting and widely researched in recent years due to the growing use of smart devices. The accent in the paper is on smartphones and smartwatches as a mean for delivering mHealth services.

The abstract is clearly written and the main contributions of the paper are clearly stated.

Section 1, introduces the main notions such as mHealth, convolutional neural networks, LSTM,  extreme learning machine,  etc.  Since there is no Related works section, the literature review is made in Section 1.  I recommend to the authors to extend the Introduction section to include other approaches to the problem. For example, in recent years Generalized Nets (GNs) have been used in the mHealth and telemedicine. An overview of the results related to  Generalized Nets models for telehealth services is presented in:

Stefanova-Pavlova, M. et al. (2017). Modeling Telehealth Services with Generalized Nets. In: Sgurev, V., Yager, R., Kacprzyk, J., Atanassov, K. (eds) Recent Contributions in Intelligent Systems. Studies in Computational Intelligence, vol 657. Springer, Cham.

Another approach is based on the Petri Nets. There are many recent works on the use of Petri nets in the field of telemedicine/telecare. For instance,  Stochastic Petri Nets are used in the modelling of wireless sensor networks in smart hospitals in the paper:

Silva FA et al., Model-Driven Impact Quantification of Energy Resource Redundancy and Server Rejuvenation on the Dependability of Medical Sensor Networks in Smart Hospitals. Sensors (Basel). 2022 Feb 18;22(4):1595.

These and other recent related works should be included in the Introduction.

Section 2 describes the Materials and Methods. It is well structured and detailed.  A structure diagram of the proposed HAR system, including the data processing unit, feature extraction unit, and classification unit is included.

 The authors should explain why 18 activities have been selected. The number of training and testing samples is sufficient.

 Subsection 2.2 on line 116 should begin on the next page.

The Categorical Cross-Entropy (CE) function and the Adam optimizer should be either explained or a reference to them should be given .

Before equation (1), the authors must include a sentence explaining the meaning of the equation. It is not accepted to start a sentence with an equation.

The structure of the  feature-extraction model is illustrated well.

Subsection 2.3 explains the Activation-Classification Model.  The sentence on line 140 must be revised as  it is unclear what the authors intended to say. 

The equation between lines 145 and 146 should not be numbered by (1) as in the previous section there is an equation. As a result, the remaining equations must be renumbered.

On line 152, it is not clear what the authors meant under “Thus, Eq. (1) could be described”.

For equation (9)-(11), an explanatory paragraph must be included as it is not clear for the non-specialists the importance of the Karush–Kuhn–Tucker conditions.

The experimental protocol is described sufficiently well. I have no remarks about it.

On line 193, the expression “measures taken in” should be replaced by “measures given by”.

The numerical results presented in Section 3, are clearly presented and the graphics confirm the correctness of the proposed approach.

The proposed by the authors approach is compared to those of other researchers in Section 4. The discussion of the results is sufficient.

The Conclusion section is too short for such a comprehensive study. It should be extended by including the possibilities of future research in this direction.

Author Response

Reviewer 2 (round 1)

Dear Anonymous Reviewer,

The authors are grateful to your comments and suggestions for improving the quality and presentation of this paper. All comments are followed. The revised parts are highlighted in red. It is our sincere hope that this revision will enhance readability and strengthen of the manuscript to satisfy the requirements of this prestigious journal.

Comments and Suggestions for Authors

The paper proposes a HAR system for classifying 18 types of physical activities using data from sensors embedded in smartphones and smartwatches. The recognition  process consists of two parts: feature extraction and HAR.

  1. The topic is interesting and widely researched in recent years due to the growing use of smart devices. The accent in the paper is on smartphones and smartwatches as a mean for delivering mHealth services.

ANS: Thanks for reviewer’s comment.

  1. The abstract is clearly written and the main contributions of the paper are clearly stated.

ANS: Thanks for reviewer’s comment.

  1. Section 1, introduces the main notions such as mHealth, convolutional neural networks, LSTM, extreme learning machine, etc.  Since there is no Related works section, the literature review is made in Section 1.  I recommend to the authors to extend the Introduction section to include other approaches to the problem. For example, in recent years Generalized Nets (GNs) have been used in the mHealth and telemedicine. An overview of the results related to Generalized Nets models for telehealth services is presented in:
  2. Stefanova-Pavlova; V. Andonov; T. Stoyanov; M. Angelova; G. Cook; B. Klein; P. Vassilev; E. Stefanova. Modeling telehealth services with generalized Nets. Recent Contributions in Intelligent Systems, 2016, 279–290.

ANS: We added a paragraph to describe the telemedicine and mHealth with deep learning and machine learning.

Line: 55-65

 In recent years, deep learning (DL) and machine learning (ML) have been widely applied in mHealth [22-25]. In these studies, DL and ML models are not only used for diagnosing, estimating, mining, and delivering physiological signals, but also for preventing chronic diseases. However, in mHealth, the big data needs to be delivered to servers, such as hospitals or health management centers. Therefore, telecommunications and navigation technologies are also important issues, which have utilized the technologies of artificial intelligence [26,27]. Stefanova-Pavlova et al. proposed the refined generalized net (GN) to track users' locations [28]. Silva et al. used Petri nets to process the reliability and availability of wireless sensor networks in the smart hospital [29]. Ruiz et al. proposed a tele-rehabilitation system to assist with physical rehabilitation during the COVID-19 pandemic [30].

  1. A. B. Said; M. F. Al-Sa’d; M. Tlili; A. A. Abdellatif; A. Mohamed; T. Elfouly; K. Harras; M. D. O’connor. A deep learning approach for vital signs compression and energy efficient delivery in mHealth systems. IEEE Access. 2018, 6, 2018, 33727-33739.
  2. A. Triantafyllidis; H. Kondylakis; D. Katehakis; A. Kouroubali; L. Koumakis; K. Marias; A. Alexiadis; K. Votis; D. Tzovaras. Deep learning in mHealth for cardiovascular disease, diabetes, and cancer: systematic review. JMIR Mhealth Uhealth, 2022, 10, e32344.
  3. T. Huang; L. Huang; R. Yang; S, Li; N. He; A. Feng; L. Li; J. Lyu. Machine learning models for predicting survival in patients with ampullary adenocarcinoma, Asia Pac J Oncol Nurs, 2022, 9, 100141.
  4. R. S. H. Istepanian; T. Al-Anzi. m-Health 2.0: new perspectives on mobile health, machine learning and big data analytics. Methods, 2018, 151, 34-40.
  5. D. M. M. Pacis; E. D. C. Subido; N. T. Bugtai. Trends in telemedicine utilizing artificial intelligence. AIP Conference Proceedings 1933, 2018, 040009.
  6. S. Bhaskar; S. Bradley; S. Sakhamuri; S. Moguilner; V. K. Chattu; S. Pandya; S. Schroeder; D. Ray; M. Banach. Designing futuristic telemedicine using artificial intelligence and robotics in the COVID-19 Era. Front. Public Health, 2020, 8, 556789.
  7. M. Stefanova-Pavlova; V. Andonov; T. Stoyanov; M. Angelova; G. Cook; B. Klein; P. Vassilev; E. Stefanova. Modeling telehealth services with generalized Nets. Recent Contributions in Intelligent Systems, 2016, 279–290.
  8. F. A. Silva; C. Brito; G. Araújo; I. Fé; M. Tyan; J.-W. Lee; T. A. Nguyen; P. R. M. Maciel. Model-driven impact quantification of energy resource redundancy and server rejuvenation on the dependability of medical sensor networks in smart hospitals. Sensors, 2022, 22, 1595.
  9. I. Ruiz; J. Contreras; J. Garcia. Towards a physical rehabilitation system using a telemedicine approach. Computer Methods in Biomechanics and Biomedical Engineering: Imaging & Visualization, 2020, 8, 671-680.

  1. Another approach is based on the Petri Nets. There are many recent works on the use of Petri nets in the field of telemedicine/telecare. For instance, Stochastic Petri Nets are used in the modelling of wireless sensor networks in smart hospitals in the paper:
  2. A. Silva; C. Brito; G. Araújo; I. Fé; M. Tyan; J.-W. Lee; T. A. Nguyen; P. R. M. Maciel. Model-driven impact quantification of energy resource redundancy and server rejuvenation on the dependability of medical sensor networks in smart hospitals. Sensors, 2022, 22, 1595.

ANS: We added a paragraph to describe the telemedicine and mHealth with deep learning and machine learning.

Line: 55-65

 In recent years, deep learning (DL) and machine learning (ML) have been widely applied in mHealth [22-25]. In these studies, DL and ML models are not only used for diagnosing, estimating, mining, and delivering physiological signals, but also for preventing chronic diseases. However, in mHealth, the big data needs to be delivered to servers, such as hospitals or health management centers. Therefore, telecommunications and navigation technologies are also important issues, which have utilized the technologies of artificial intelligence [26,27]. Stefanova-Pavlova et al. proposed the refined generalized net (GN) to track users' locations [28]. Silva et al. used Petri nets to process the reliability and availability of wireless sensor networks in the smart hospital [29]. Ruiz et al. proposed a tele-rehabilitation system to assist with physical rehabilitation during the COVID-19 pandemic [30].

  1. These and other recent related works should be included in the Introduction.

ANS: We added a paragraph to describe the telemedicine and mHealth with deep learning and machine learning.

Line: 55-65

 In recent years, deep learning (DL) and machine learning (ML) have been widely applied in mHealth [22-25]. In these studies, DL and ML models are not only used for diagnosing, estimating, mining, and delivering physiological signals, but also for preventing chronic diseases. However, in mHealth, the big data needs to be delivered to servers, such as hospitals or health management centers. Therefore, telecommunications and navigation technologies are also important issues, which have utilized the technologies of artificial intelligence [26,27]. Stefanova-Pavlova et al. proposed the refined generalized net (GN) to track users' locations [28]. Silva et al. used Petri nets to process the reliability and availability of wireless sensor networks in the smart hospital [29]. Ruiz et al. proposed a tele-rehabilitation system to assist with physical rehabilitation during the COVID-19 pandemic [30].

  1. A. B. Said; M. F. Al-Sa’d; M. Tlili; A. A. Abdellatif; A. Mohamed; T. Elfouly; K. Harras; M. D. O’connor. A deep learning approach for vital signs compression and energy efficient delivery in mHealth systems. IEEE Access. 2018, 6, 2018, 33727-33739.
  2. A. Triantafyllidis; H. Kondylakis; D. Katehakis; A. Kouroubali; L. Koumakis; K. Marias; A. Alexiadis; K. Votis; D. Tzovaras. Deep learning in mHealth for cardiovascular disease, diabetes, and cancer: systematic review. JMIR Mhealth Uhealth, 2022, 10, e32344.
  3. T. Huang; L. Huang; R. Yang; S, Li; N. He; A. Feng; L. Li; J. Lyu. Machine learning models for predicting survival in patients with ampullary adenocarcinoma, Asia Pac J Oncol Nurs, 2022, 9, 100141.
  4. R. S. H. Istepanian; T. Al-Anzi. m-Health 2.0: new perspectives on mobile health, machine learning and big data analytics. Methods, 2018, 151, 34-40.
  5. D. M. M. Pacis; E. D. C. Subido; N. T. Bugtai. Trends in telemedicine utilizing artificial intelligence. AIP Conference Proceedings 1933, 2018, 040009.
  6. S. Bhaskar; S. Bradley; S. Sakhamuri; S. Moguilner; V. K. Chattu; S. Pandya; S. Schroeder; D. Ray; M. Banach. Designing futuristic telemedicine using artificial intelligence and robotics in the COVID-19 Era. Front. Public Health, 2020, 8, 556789.
  7. M. Stefanova-Pavlova; V. Andonov; T. Stoyanov; M. Angelova; G. Cook; B. Klein; P. Vassilev; E. Stefanova. Modeling telehealth services with generalized Nets. Recent Contributions in Intelligent Systems, 2016, 279–290.
  8. F. A. Silva; C. Brito; G. Araújo; I. Fé; M. Tyan; J.-W. Lee; T. A. Nguyen; P. R. M. Maciel. Model-driven impact quantification of energy resource redundancy and server rejuvenation on the dependability of medical sensor networks in smart hospitals. Sensors, 2022, 22, 1595.
  9. I. Ruiz; J. Contreras; J. Garcia. Towards a physical rehabilitation system using a telemedicine approach. Computer Methods in Biomechanics and Biomedical Engineering: Imaging & Visualization, 2020, 8, 671-680.

6.Section 2 describes the Materials and Methods. It is well structured and detailed.  A structure diagram of the proposed HAR system, including the data processing unit, feature extraction unit, and classification unit is included.

ANS: Many thanks for reviewer’s comment.

7.The authors should explain why 18 activities have been selected. The number of training and testing samples is sufficient.

ANS: We modified this paragraph and added one reference in 2.1 sector.

Line: 113-128

2.1 UCI-WIDSM Dataset

The UCI-WISDM dataset [40] is comprised of tri-axial accelerometer and gyroscope data obtained from 51 volunteer subjects. The subjects carried an Android phone (Google Nexus 5/5x or Samsung Galaxy S5) in the front pockets of their pants and an Android watch (LG G Watch) at their wrist while performing eighteen activities, which were categorized as body movements (walking, jogging, walking stairs, sitting, and standing ) included in many previous studies, hand movements (kicking, dribbling, catching, typing, writing, clapping, brushing teeth, and folding clothes) representing the activities of daily life, and eating movements (eating pasta, drinking soup, eating sandwich, eating chips, and drinking from cup) investigating the feasibility of automatic food tracking applications [41]. The data was sampled at a rate of 20 Hz, and the 12 signals were segmented into fixed-width sliding windows of 6.4 seconds with 50% overlap between them. Each sample contained 12-channel signals, and each channel comprised of 128 points. Samples containing two activities were removed. The numbers of training and testing samples were 34,316 and 14,707, respectively, and the sample numbers for each of the eighteen activities are presented in Table 1.

  1. G. M. Weiss; K. Yoneda; T. Hayajneh. Smartphone and smartwatch-based biometrics using activities of daily living. IEEE Access, 2019, 7, 133190-133202.

  1. Subsection 2.2 on line 116 should begin on the next page.

ANS: We modified the typesetting of this paragraph.

  1. The Categorical Cross-Entropy (CE) function and the Adam optimizer should be either explained or a reference to them should be given .

ANS: We added one reference.

Line: 148-150

The loss function is the categorical Cross-Entropy (CE) function, and the Adam optimizer is used [42], with the learning rate set to 0.0001. Eq. (1) is the formular of categorical CE,

  1. I. Goodfellow; Y. Bengio; A. Courville. Deep learning. MIT Press, 18 Nov. 2016.

  1. Before equation (1), the authors must include a sentence explaining the meaning of the equation. It is not accepted to start a sentence with an equation.

ANS: We modified the sentences.

Line: 148-152

The loss function is the categorical Cross-Entropy (CE) function, and the Adam optimizer is used [42], with the learning rate set to 0.0001. Eq. (1) is the formula of categorical CE,

(1)

where M is 18, ak is the score of sofmax for the positive class, and ai is the scores inferred by the net for each class.

  1. The structure of the feature-extraction model is illustrated well.

ANS: Many thanks for reviewer’s comment.

  1. Subsection 2.3 explains the Activation-Classification Model. The sentence on line 140 must be revised as it is unclear what the authors intended to say.

ANS: We modified the sentences.

Line: 155-161

2.3 Activation-Classification Model

The activation-classification model was a single layer feedforward neural (SLFN) network with the ELM algorithm [43]. Its advantages are the convergent time being shorter than BP method, not converging to the local minimum. For a SLFN, a training set S = {(Xr, Yir| Xi = (xr1, xr2, …, xrn)T ∈ Rn, Yr= (yr1, yr2, …, yrm)T ∈ Rm}, where Xr denotes the rth input vector and Yr represents the rth target vector, the output o of SLFN with l hidden neurons can be expressed as:

  1. The equation between lines 145 and 146 should not be numbered by (1) as in the previous section there is an equation. As a result, the remaining equations must be renumbered.

ANS: We corrected this mistake.

Line: 161

(2)

  1. On line 152, it is not clear what the authors meant under “Thus, Eq. (1) could be described”.

ANS: We modified the sentences.

Line: 168-169

The output o of SLFN is equal to the target output y. Thus, Eq. (2) could be described,

(4)

Y=Hβ,

(5)

  1. For equation (9)-(11), an explanatory paragraph must be included as it is not clear for the non-specialists the importance of the Karush–Kuhn–Tucker conditions.

ANS: Because RELM was proposed by Den et al. In ref [45], RELM was detailly described. Thus, we only described how to get α, β, and ε by the Lagrange multiplier. Moreover, we added one reference to explain the meaning of KKT optimality condition.

Line: 180-184

The method of Lagrange multipliers is used to search the optimal solution of Eq. (8),

(9)

where α is the Lagrange multiplier with the equality constraints of Eq. (9). Setting the gradients of L(β ,ε ,α ) equal to zero gives the following Karush–Kuhn–Tucker (KKT) optimality conditions [45,46].

,

(10)

.

(11)

.

(12)

  1. W. Deng ; Q. Zheng ; L. Chen. Regularized extreme learning machine. 2009 IEEE Symposium on Computational Intelligence and Data Mining, 2009, 389-395.
  2. https://en.wikipedia.org/wiki/Karush%E2%80%93Kuhn%E2%80%93Tucker_condition

  1. The experimental protocol is described sufficiently well. I have no remarks about it.

ANS: Many thanks for reviewer’s comment.

  1. On line 193, the expression “measures taken in” should be replaced by “measures given by”.

ANS: We modified this sentence.

Line: 209-210

In this work, the performance of the proposed method was evaluated using the measures given by Equations (13)–(16):

  1. The numerical results presented in Section 3, are clearly presented and the graphics confirm the correctness of the proposed approach.

ANS: Many thanks for reviewer’s comment.

  1. The proposed by the authors approach is compared to those of other researchers in Section 4. The discussion of the results is sufficient.

ANS: Many thanks for reviewer’s comment. But, according to the comment of reviewer 1, We modified the texts in Table 8 and added the information about these previous studies classifying the number of activities and using wearable devices.

Line: 348-355

Table 8 presents a comparative analysis of our proposed method with other studies that utilized the UCI-WISDM Smartphone and / or Smartwatch Activity and Biometrics dataset for six / eighteen activities. Previous studies [27,37,38,39,40,41] only classified six activities, while studies [28,29] classified eighteen activities. As shown, the proposed HAR system using the hybrid CNN+BiGRU model and SLFN with RELM achieved an F1-score and accuracy of 98.4% and 98.2%, respectively, which is among the best results reported in the literature.

Table 8 Comparative result of various methods using UCI-WISDM dataset.

Ref.

Classification Method

Activities/

Wearable Devices

F1-Score (%)

Accuracy (%)

[27]

CNN+GRU

6 / Phone

91.7

NA

[28]

Riege Forest

18 / Phone & Watch

NA

94.4

[29]

CNN+LSTM

18 / Watch

96.3

96.2

[37]

CNN+ Handcrafted Features

6 / Phone

NA

93.3

[38]

ConvAS

6 / Phone

NA

94.9

[39]

CNN+LSTM

6 / Phone & Watch

NA

96.0

[40]

Hesitant Fuzzy Belief

Structures

6 / Phone & Watch

NA

95.82

[41]

ConvAE-LSTM

6 / Phone

97.4

97.1

Proposed method

Hybrid CNN+BGRU

SLFN with RELM

18 / Phone & Watch

98.4

98.2

20.The Conclusion section is too short for such a comprehensive study. It should be extended by including the possibilities of future research in this direction.

ANS: We modified the texts of Conclusions.

Line: 368-381

  1. Conclusions

The proposed deep learning model utilizes the hybrid CNN+BiGRU for feature extraction from the signals of sensors embedded on the smartphone and smartwatch, and the SLFN with the RELM algorithm for the classification of 18 physical activities, including body, hand, and eating movements. The experimental results demonstrate that the proposed model outperforms other existing schemes that utilize deep learning or machine learning methods in terms of F1-score and accuracy. Notably, the worst F1-score was found in the classification for brushing teeth. Our investigation shows that using different deep learning models for feature extraction and classification during the training phase can effectively increase recognition accuracy and training time. Moreover, since the data is recorded from the smartphone and smartwatch, our proposed method has the potential to be used for mHealth in real-time at the environments without embedding the wireless sensor networks. The weakness of this study is to ignore the signals when two activities are transferring. Thus, in the future, we will explore this problem.

Round 2

Reviewer 2 Report

The has been significantly improved and the authors have taken into account all of my remarks.